# Consensus-Based SOC Balancing of Battery Energy Storage Systems in Wind Farm

**Cao-Khang Nguyen [1], Thai-Thanh Nguyen [1], Hyeong-Jun Yoo [1] and Hak-Man Kim [1,2,*]**

[1]  Department of Electrical Engineering, Incheon National University, Songdo-dong, 119 Academy-ro, Yeonsu-gu, Incheon 22012, Korea; khang.nguyencao@gmail.com (C.-K.N.); ntthanh@inu.ac.kr (T.-T.N.); yoohj@inu.ac.kr (H.-J.Y.)
[2]  Research Institute for Northeast Asian Super Grid, Incheon National University, 119 Academy-ro, Yeonsu-gu, Incheon 22012, Korea
*  Correspondence: hmkim@inu.ac.kr; Tel.: +82-32-835-8769; Fax: +82-32-835-0773

**Abstract:** Multiple battery energy storage systems (BESSs) are used to compensate for the fluctuation in wind generations effectively. The stage of charge (SOC) of BESSs might be unbalanced due to the difference of wind speed, initial SOCs, line impedances and capabilities of BESSs, which have a negative impact on the operation of the wind farm. This paper proposes a distributed control of the wind energy conversion system (WECS) based on dynamic average consensus algorithm to balance the SOC of the BESSs in a wind farm. There are three controllers in the WECS with integrated BESS, including a machine-side controller (MSC), the grid-side controller (GSC) and battery-side controller (BSC). The MSC regulates the generator speed to capture maximum wind power. Since the BSC maintains the DC link voltage of the back-to-back (BTB) converter that is used in the WECS, an improved virtual synchronous generator (VSG) based on consensus algorithm is used for the GSC to control the output power of the WECS. The functionalities of the improved VSG are designed to compensate for the wind power fluctuation and imbalance of SOC among BESSs. The average value of SOCs obtained by the dynamic consensus algorithm is used to adjust the wind power output for balancing the SOC of batteries. With the proposed controller, the fluctuation in the output power of wind generation is reduced, and the SOCs of BESSs are maintained equally. The effectiveness of the proposed control strategy is validated through the simulation by using a MATLAB/Simulink environment.

**Keywords:** wind energy conversion system; distributed control; battery energy storage system; consensus algorithm

## 1. Introduction

Renewable energy sources (RESs) such as wind power and solar energy have been widely integrated into the power system [1–3]. One of the fastest growing RESs is wind power, which has received more and more attention from researchers in recent years [4]. However, the uncertainty of wind power results in the fluctuation in system frequency. Therefore, an increase in the penetration of wind power might cause several adverse effects to the power quality, stability and reliability of the power system [5,6]. In order to mitigate the impacts of wind power fluctuation, battery energy storage system (BESS) has been integrated into the wind energy conversion system (WECS) [7,8]. By controlling the charging/discharging power of the battery, the output power of the WECS can be smoother. For the large-scale wind farm, multiple BESSs could be used for addressing the problem of wind power fluctuation [9].

The imbalance of state of charge (SOC) is a common issue in the multi-cell battery systems due to the manufacturing variance, internal impedance, self-discharge rates, etc. [10,11]. The SOC imbalance might result in the reduction of charge capacity, early termination of charging or discharging, and accelerated battery degradation [10]. This problem is also observed in the microgrid or wind farm systems with multiple BESSs that are operated independently. Several studies have addressed the issue of SOC imbalance in the independent BESSs [12–22]. The SOC of the BESSs is unbalanced due to the difference of wind power output, initial SOC, capacities of BESSs, and line impedances [12]. The batteries with the lowest SOC might stop the operation of the WECS because it does not have enough power to support the system. The remaining batteries should be in charge of compensating power for the load change, which results in the accelerated charge or discharge rate of BESSs and the reduction of service life of BESSs [13]. The system could be collapsed if the energy storage of the remaining BESSs is not enough to compensate for the disturbance [14]. When multiple BESSs are used in the wind farm, the system reliability can be improved by maintaining the state of charges (SOC) balancing of BESSs [12].

In order to deal with the unbalanced SOC problem, several control strategies for SOC balancing have been introduced [15–22]. The SOC-based droop control strategies of the converter have been used widely to achieve proper power sharing and SOC balancing. The authors in reference [15] proposed a double-quadrant SOC-based droop control to achieve an appropriate power sharing in the autonomous DC microgrid. The SOC-based adaptive droop control was introduced for balancing the SOCs among the energy storage systems (ESU) in DC microgrid [16]. The paper in [17] proposed a multifunctional and wireless droop control to eliminate the SOC imbalance of the ESUs based on the $P - f$ droop controller. However, the SOC-based droop control strategies cause the frequency deviation from the nominal values, which affects to the performance of the system.

Instead of using the SOC-based droop control, several SOC balancing control schemes based on the centralized control were proposed to balance the discharge rate of the energy storage system [18,19]. A coordinated secondary control based on current sharing control was introduced to achieve the discharge rate balancing and avoid the overcurrent among distributed generations in the islanded AC microgrid. However, the use of the centralized controller for SOC balancing might reduce the system reliability since all converters rely on the centralized controller [20]. The SOC balancing controllers based on decentralized control or distributed control have been proposed recently to improve the problem of SOC imbalance and enhance the system reliability [21,22]. In reference [21], a multi-agent based SOC balancing control was discussed, while in reference [22], the SOC balancing control was achieved by using coordinated secondary control to adjust virtual impedance loops.

However, most of the papers in the literature mainly focused on the stand-alone system. In addition, the impact of wind generation on the SOC of batteries was neglected. In this paper, an improved virtual synchronous generator (VSG) control based on consensus algorithm is proposed to regulate the output power of wind energy conversion system, which considers the impacts in wind speed variation on the WECS. The proposed controller consists of two main parts, including a consensus based SOC balancing controller and smoothing power controller. With the use of consensus-based control algorithm, the SOC of batteries is controlled to converge at the same value. The batteries which have higher SOC will inject more power to support the batteries with lower SOC. In addition, the low-pass filter is added to the VSG control of grid-side converter for smoothing active output power of the WECS. Hence, the performance of the WECS can be improved, and the system stability can be enhanced.

The paper is arranged as follows: Section 2 presents the overall structure of the WECS and the detailed control diagram of power converters in the WECS. The proposed control strategy based on consensus algorithm is introduced in Section 3. The performance of the proposed control strategy is validated through the simulation results in Section 4. Finally, the conclusion appears in Section 5.

## 2. Structure of Wind Energy Conversion System with Integrated Battery Energy Storage System

### 2.1. Overall Structure

The overall structure of the WECS with integrated BESS is illustrated in Figure 1. The wind turbine is directly connected to the permanent magnet synchronous generator (PMSG). The PMSG is interfaced with the utility grid through the back-to-back (BTB) converter, which includes three main parts, namely, machine-side converter (MSC), grid-side converter (GSC) and battery-side converter (BSC). The MSC controls the generator in order to capture maximum active power from wind turbine by using maximum power point tracking (MPPT) algorithm. In the BSC, a bidirectional DC-DC converter is utilized to maintain the constant DC capacitor voltage of the BTB converter by charging or discharging power. With the use of BESS, instead of controlling the DC capacitor voltage, the GSC is flexible to adjust the output power and support the system frequency. The power converter used in the GSC can be classified into two types, grid-feeding converter and grid-forming converter [23]. In this structure, the grid-forming converter is adopted to regulate the active power and support the system frequency by using the VSG control. The advantages of VSG control for supporting system frequency by imitating both steady-state and transient-state characteristics of the synchronous generator are explicitly discussed in previous works [24,25]. Therefore, it will not be further considered in this paper. The detailed control diagram of each converter in the WECS is analyzed in the next part.

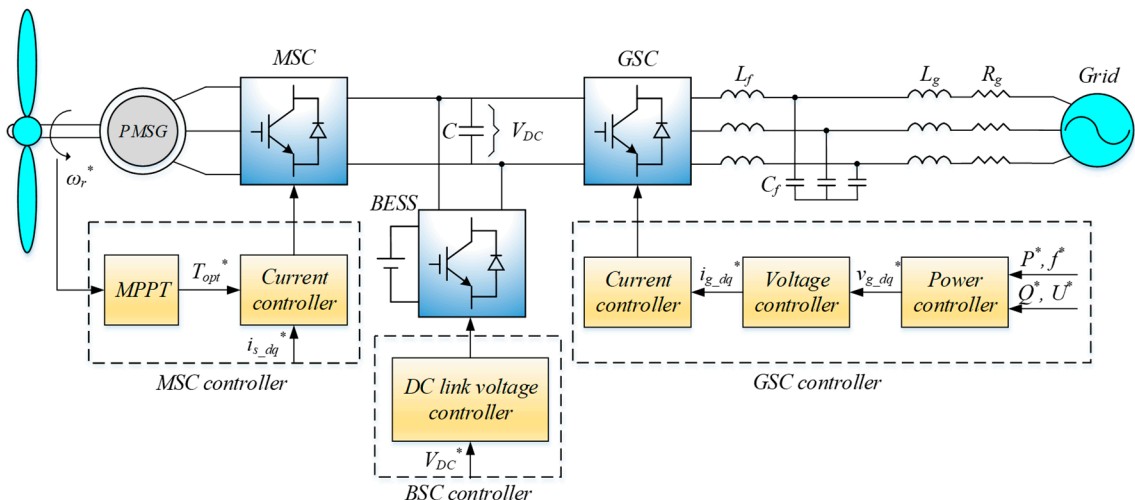

**Figure 1.** Overall structure of the wind energy conversion system (WECS) integrated with battery energy storage system (BESS).

### 2.2. Control of a Machine-Side Converter

The machine-side converter is utilized to control the generator of the WECS. Depending on the variation of wind speed, the rotor speed is varied, and thus, the output power generated by the PMSG fluctuates. By applying the MPPT algorithm, electromagnetic torque is regulated to be proportional to the square of rotor speed so that the optimal value is maintained. Therefore, with particular wind speed, the wind turbine can produce maximum possible power. The control scheme of MSC is shown in Figure 2a, which includes two cascaded controllers, namely torque controller, and current controller. With regard to the torque controller, the reference of electromagnetic torque is achieved by the MPPT algorithm, as shown below:

$$T_e^* = -k_{opt}\omega_r^2 \tag{1}$$

where $T_e^*$ is the optimal torque generated from wind turbine, $\omega_r$ is the speed of rotor, $k_{opt}$ is the optimal coefficient wind turbine, which is described as:

$$k_{opt} = \frac{0.5\rho A r^3 C_{pmax}}{\lambda_{opt}^3}\omega_r^3 \tag{2}$$

where $A$ is the turbine swept area, $r$ is the turbine radius, $\rho$ is the air mass density, $\lambda_{opt}$ is the optimal tip-speed ratio when the blade pitch angle $\beta = 0$, $C_{pmax}$ is the maximum performance coefficient.

The optimal torque determined by the torque controller is used to calculate the current reference of the current controller, as given by:

$$i_{sq}^* = \frac{2T_e^*}{3p\lambda_m} \tag{3}$$

where $\lambda_m$ is the flux generated from the permanent magnet, $i_{sq}^*$ is the reference of stator current in $q$-axis, $p$ is the number of pole pairs in the PMSG.

While the MPPT algorithm is applied to the torque control loop, the current controller adjusts the stator current of the generator in $dq$ reference frame. The current in d-axis is controlled at zero to maintain the linearization between electrical torque and the q-axis current, as shown in Equation (3). The output voltage reference of the generator can be determined through the inner current loop, which is given by:

$$v_d^* = k_{pm}(i_{sd}^* - i_{sd}) + k_{im}\int (i_{sd}^* - i_{sd})dt - \omega_r L_q i_{sq} \tag{4}$$

$$v_q^* = k_{pm}\left(i_{sq}^* - i_{sq}\right) + k_{im}\int \left(i_{sq}^* - i_{sq}\right)dt + \omega_r L_d i_{sd} + \omega_r \lambda_m \tag{5}$$

where $i_{sd}$ and $i_{sq}$ are the stator current in $dq$ reference frame, $v_d^*$ and $v_q^*$ are the reference of stator voltage of the PMSG and $k_{pm}$ and $k_{im}$ are the PI parameters of the current controller in the MSC.

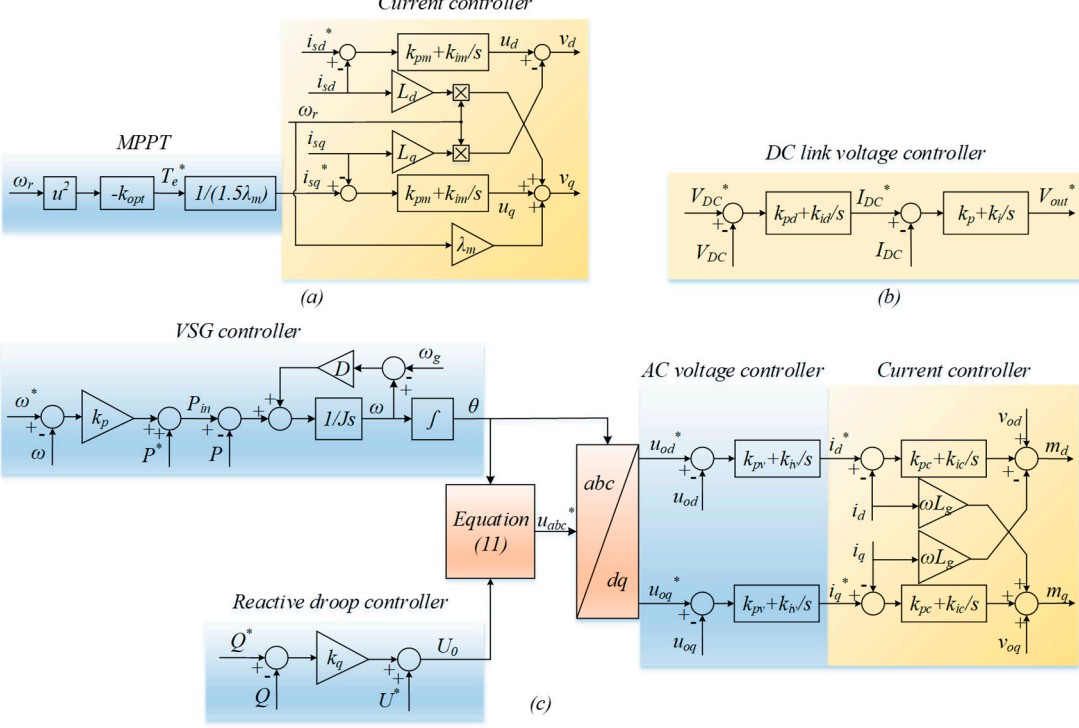

**Figure 2.** Control diagram of power converter in the WECS: (**a**) Machine-side converter, (**b**) Battery-side converter, (**c**) Grid-side converter.

### 2.3. Control of a Battery-Side Converter

The BSC adopts the bi-directional DC-DC converter to regulate the DC capacitor voltage of the BTB converter. In addition, it plays an important role in balancing the power between the GSC and the MSC. When the wind power is lower than the power required from the grid, the power from the BSC is released. By contrast, when the power generated from a wind turbine is larger than the power requirement from the grid, the power is reserved by the BSC. The detailed control scheme of the BSC is shown in Figure 2b. Whereas the outer control is designed to maintain the constant DC capacitor voltage of the BTB converter, the inner controller adjusts the current flowing to the inductor of the bi-directional DC-DC converter. The output voltage and current controller of the converter can be calculated by these equations:

$$I_{DC}^* = k_{pd}(V_{DC}^* - V_{DC}) + k_{id} \int (V_{DC}^* - V_{DC})dt \tag{6}$$

$$V_{out}^* = k_p(I_{DC}^* - I_{DC}) + k_i \int (I_{DC}^* - I_{DC})dt \tag{7}$$

where $k_{pd}$ and $k_{id}$ are the PI parameters of the DC capacitor voltage controller, $k_p$ and $k_i$ are the PI parameters of the current controller in the bidirectional DC-DC converter and $V_{out}^*$ is the output voltage reference.

The design of the PI parameters of voltage and current controller should consider the different bandwidth in each control loop. The current controller is designed for a bandwidth of 1.6 kHz with good rejection of high-frequency disturbance, whereas the voltage controller is designed for 400 Hz bandwidth to ensure the stability of the converter system [26]. The classical pole-zero and bode techniques are used to select these parameters.

### 2.4. Control of a Grid-Side Converter

With regard to the grid-side converter, its detailed control diagram is shown in Figure 2c, which consists of three cascaded control loops: Power controller, voltage controller and current controller. In the power controller, virtual synchronous generator control is applied to active power control to regulate the power angle $\theta$ while reactive power control adjusts the AC voltage reference by using droop controller. With the use of VSG control, the GSC can imitate both steady-state and transient-state characteristics of a synchronous generator to virtually provide inertia to the WECS. The VSG control is described by the swing equation, which is given by:

$$P_{in} - P = J\omega \frac{d\omega}{dt} + D(\omega - \omega_g) \tag{8}$$

where $P$ is the measured active power of the GSC, $J$ is the virtual inertia coefficient, $\omega$ is the rotor angular frequency generated by VSG control, $\omega_g$ is the grid frequency and $P_{in}$ is the input power which is determined by the governor:

$$P_{in} = P^* + k_p(\omega^* - \omega) \tag{9}$$

where $P^*$ the set active power, $k_p$ is the droop coefficient for active power controller and $\omega^*$ is the nominal angular frequency.

Regarding the reactive power controller, the amplitude of the output voltage reference in *abc* reference frame can be calculated by the droop controller, which is given as:

$$U_0 = U^* + k_q(Q^* - Q) \tag{10}$$

where $U_0$ is the amplitude of output AC voltage, $k_q$ is the droop coefficient for reactive power controller and $Q^*$ is the set reactive power.

The reference of output voltage for the GSC is determined by combining Equations (9) and (10), which is shown as follows:

$$\begin{cases} u_a^* = U_0\sin\theta \\ u_b^* = U_0\sin(\theta - 2\pi/3) \\ u_c^* = U_0\sin(\theta + 2\pi/3) \end{cases} \tag{11}$$

The reference of the output AC voltage is transferred from *abc* reference frame to *dq* reference frame, and then input to the inner control loop. The inner control loop consists of two controllers, namely voltage controller and current controller. The voltage controller is used to maintain the output voltage of the GSC, which is described as follows:

$$i_{ld}^* = k_{pv}(v_{od}^* - v_{od}) + k_{iv}\int (v_{od}^* - v_{od})dt \tag{12}$$

$$i_{lq}^* = k_{pv}\left(v_{oq}^* - v_{oq}\right) + k_{iv}\int \left(v_{oq}^* - v_{oq}\right)dt \tag{13}$$

where $k_{pv}$ and $k_{iv}$ are the proportional and integral gain of voltage controller, respectively.

Regarding the current controller, it is employed to control the inductor current of the GSC. The current reference is also the output of the voltage controller. Similar to the voltage controller, the reference value of the current controller is also compared with the measured current and then input to the PI controller. The output of the current controller can be treated by:

$$m_d = v_{od} - \omega i_q L_f + k_{pc}(i_{ld}^* - i_{ld}) + k_{ic}\int (i_{ld}^* - i_{ld})dt \tag{14}$$

$$m_q = v_{oq} + \omega i_d L_f + k_{pc}\left(i_{lq}^* - i_{lq}\right) + k_{ic}\int \left(i_{lq}^* - i_{lq}\right)dt \tag{15}$$

where $k_{iv}$ and $k_{iv}$ are the proportional and integral component of the current controller, respectively, and $m_d$ and $m_q$ are the modulating signals in the *dq* reference frame.

## 3. Proposed Control Strategy

In the proposed control method, a consensus algorithm is applied for achieving SOC balance in the WECS while the low pass filter is added to compensate the power fluctuation of the WECS. The average dynamic consensus algorithm was introduced in reference [27], which is presented by:

$$\dot{\zeta}_i = \sum_{j=1}^{N} a_{ij}\left(\zeta_i - \zeta_j\right) \qquad i = 1, 2, \ldots, N \tag{16}$$

where $\zeta_i$ and $\zeta_j$ are the state variables for node *i* and *j*, respectively, $a_{ij}$ is the weight for exchanging information from node *j* to *i*.

The communication network is described by a graph with nodes N and edges . Each node of the graph represents an agent and each edge of edges represents a communication link between two particular agents. The agents communicate with their neighbors to exchange information among them, and then update the state information. The Laplacian matrix $L_N = [l_{ij}]\epsilon R^{n\times n}$ associated with the graph can be written as:

$$\begin{cases} l_{ii} = \sum_{j=1,\ j\neq i}^{N} a_{ij} \\ l_{ij} = -a_{ij}\, j \neq i \end{cases} \tag{17}$$

Therefore, the dynamics of the consensus algorithm can be described by:

$$\dot{X} = -L_N X \tag{18}$$

where $X = [x_1, x_2, \ldots, x_n]^T$ is the state variable vector.

The SOC of batteries among WECSs are considered as the state variables of the consensus algorithm. Through the consensus algorithm, the reference of SOC is updated after each time interval. The current SOC of battery $i$ can be calculated as:

$$SOC_i = SOC_i^* - \frac{1}{C_{bati}} \int_0^t I_i dt \tag{19}$$

where $SOC_i^*$ and $SOC_i$ are the initial SOC and the current state of charge of battery $i$, $I_i$ and $C_{bati}$ are the output current and the capacity of battery $i$.

The power generated from the SOC controller can be calculated by:

$$P_{SOC} = k_{SOC}(SOC^* - SOC_i) = k_{SOC}\Delta SOC_i \tag{20}$$

where $k_{SOC}$ is the SOC coefficient, $SOC^*$ is the SOC reference updated from the consensus algorithm and $\Delta SOC_i$ is the difference between the updated SOC reference and the current SOC of battery $i$.

A low-pass filter is used to smoothen the fluctuation of the output wind power, and then the subtraction of filtered wind power from the measured wind power is used as the compensated power for the fluctuation. Since each WECS has the ability to smoothen wind power, the total power of all WECSs transferred to the grid could be smoothed. The reference of output power is added to regulate the output power of the WECS, as shown in Figure 3.

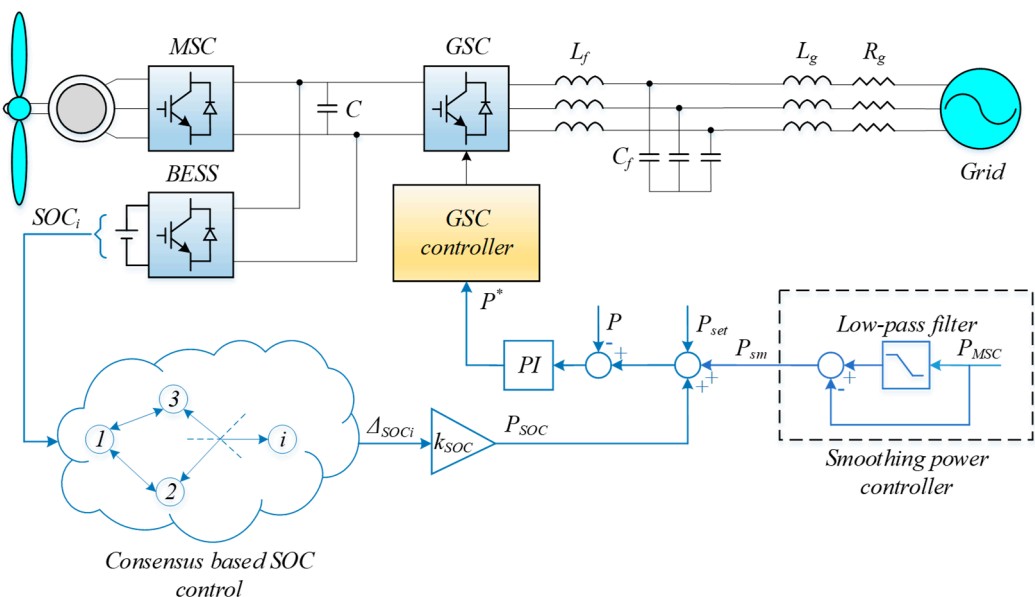

**Figure 3.** Control diagram of the proposed control strategy.

## 4. Simulation Results

In order to evaluate the efficiency of the proposed control strategy, a simulation system is carried out by using a SimPowerSystems toolbox in a MATLAB/Simulink environment. The detailed simulation system which consists of three WECSs with integrated BESS is presented in Figure 4. These WECSs are connected in parallel through the impedance lines and then interfaced with the utility grid. Each WECS injects 1.5 (MW) active power to the utility grid. The proposed controller is verified through two case studies. Whereas the performance of the system is analyzed with different initial SOCs in case study 1, the same initial SOC is focused on case study 2. The simulation parameters of PMSG are listed in Table 1.

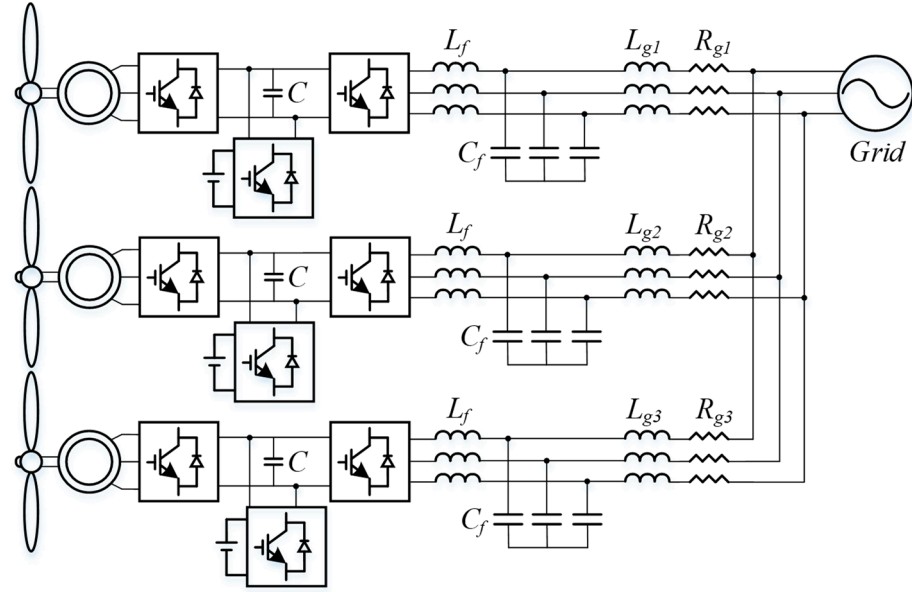

**Figure 4.** Simulation system.

**Table 1.** System parameters.

| Parameters | Values | Parameters | Values |
|---|---|---|---|
| Rated power | 2.45 (MW) | Number of pole pairs | 8 |
| Rated line-to-line voltage | 4000 (V) | Synchronous inductance | 9.816 (mH) |
| Rated stator current | 490 (A) | Rated rotor speed | 400 (rpm) |
| Rated stator frequency | 53.3 (Hz) | Flux | 7.03 (Wb) |
| Active power droop coefficient | $4 \times 10^{-5}$ | Reactive power droop coefficient | 0.068 |

### 4.1. Case Study 1: Performance of the Proposed Control Strategy with Different Initial SOCs

In the first case, the average SOC of batteries is chosen to equal 45%, 35%, and 25% at the beginning, respectively. The wind speed flowing to three wind turbines is chosen differently for each one, as shown in Figure 5. Whereas the average wind speed of WECS1 is the highest, WECS3 receives the lowest average wind speed. The variation of wind speed results in the fluctuation of active power generated from the wind turbine, which is measured through the MSC, as shown in Figure 6. Although the output power transferred to the utility grid is regulated at 1.5 (MW) for each WECS, the power captured from the wind turbine is not enough for satisfying the power requirement of the utility grid. Therefore, the batteries discharge power in order to achieve power balance between the MSC and the GSC and compensate the power fluctuation.

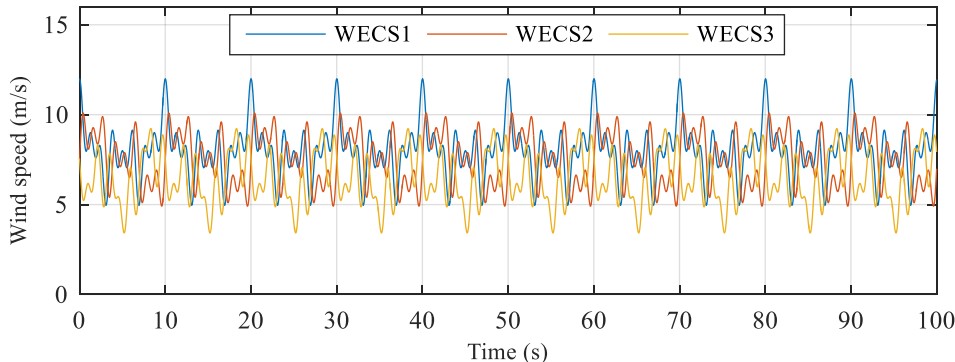

**Figure 5.** Wind speed.

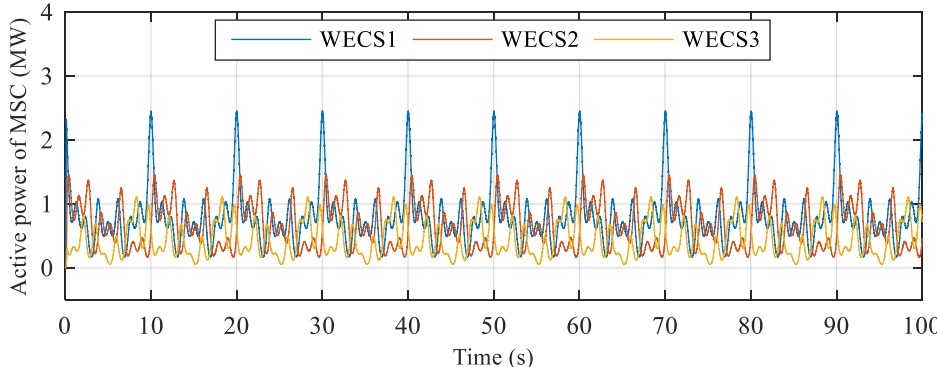

**Figure 6.** Active power of machine-side converter.

It can be seen from Figure 7 that, without the proposed control strategy, the SOC of each battery moves down slightly at the first interval of 50 s. After that, whereas the SOC of battery 1 and battery 2 are higher than 20%, the SOC of battery 3 moves down to under 20%. However, when the consensus algorithm is applied, the SOC of battery 1 moves down quickly while that of battery 3 moves up in the first 30 s, as shown in Figure 8. In other words, battery 1, which has the highest SOC, releases more power to support the other batteries. Consequently, battery 2 discharges less power and battery 3 reserves power because it has the lowest SOC. After 50 s, all three batteries discharge power to the GSC with the same SOC. In addition, none of the batteries have SOC below 20%. Hence, the proposed controller can reduce the adverse effects of the low SOC on the DC capacitor voltage and the WECS, which has the lowest SOC, can survive.

Furthermore, Figure 9 presents the active power of the GSC without the proposed controller. Because the SOC of batteries is not regulated, the active output power of the GSC is maintained equally at 1.5 MW. However, as illustrated in Figure 10, when the proposed controller is applied, WECS1, which has the highest SOC, injects more power than WECS2 and WECS3 until the SOCs among the WECSs reach the same value at roughly 50 s. After the SOCs are balanced, the output power exchanged with the utility grid is maintained at the reference value. In addition, with the use of smoothing power controller, the fluctuation of wind power is compensated by the battery of each WECS, as shown in Figures 11 and 12. Therefore, the output power of WECSs can be smooth, which enhances the power quality of the system.

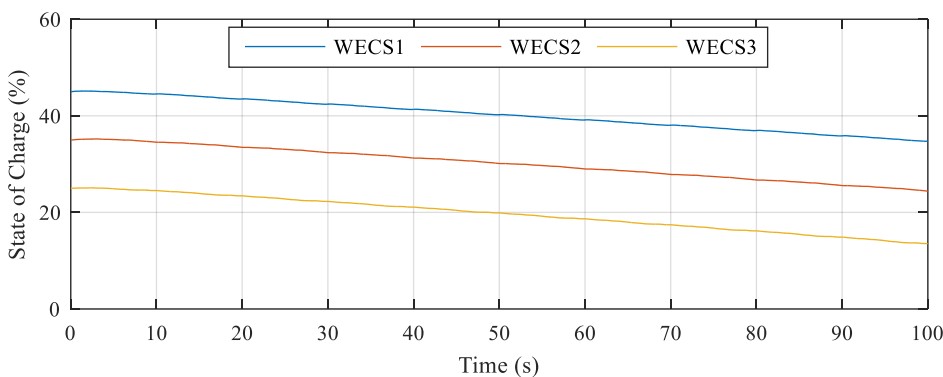

**Figure 7.** State of charge (SOC) of batteries in the WECS without proposed controller (Case 1).

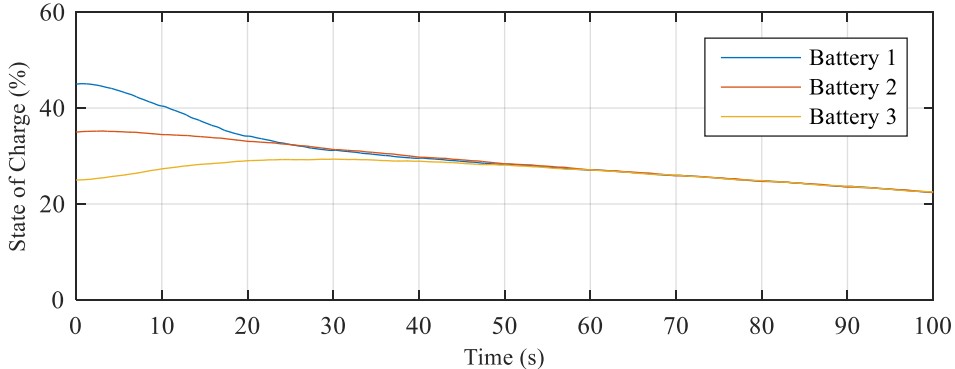

**Figure 8.** State of charge of batteries in the WECS with proposed controller (Case 1).

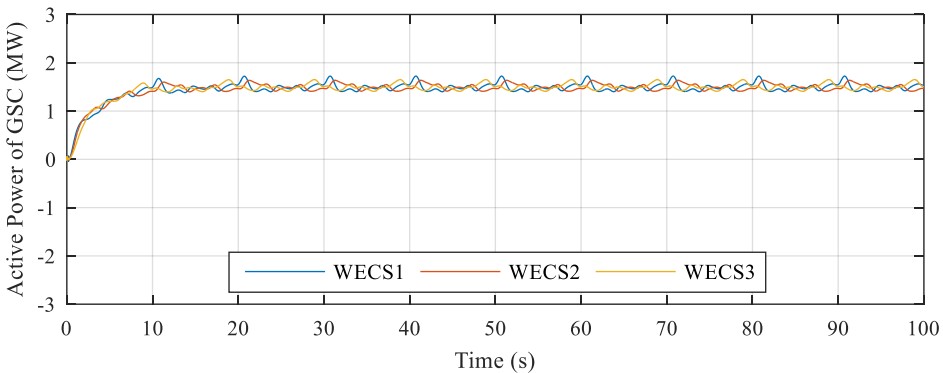

**Figure 9.** Active power of grid-side converter (GSC) without the proposed controller (Case 1).

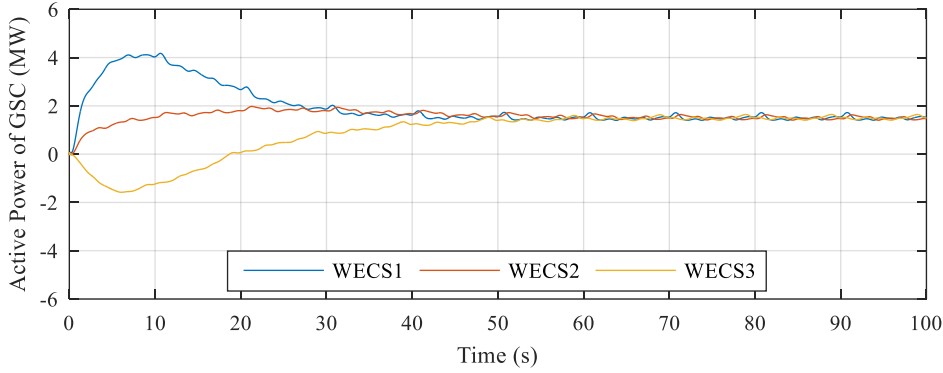

**Figure 10.** Active power of GSC with the proposed controller (Case 1).

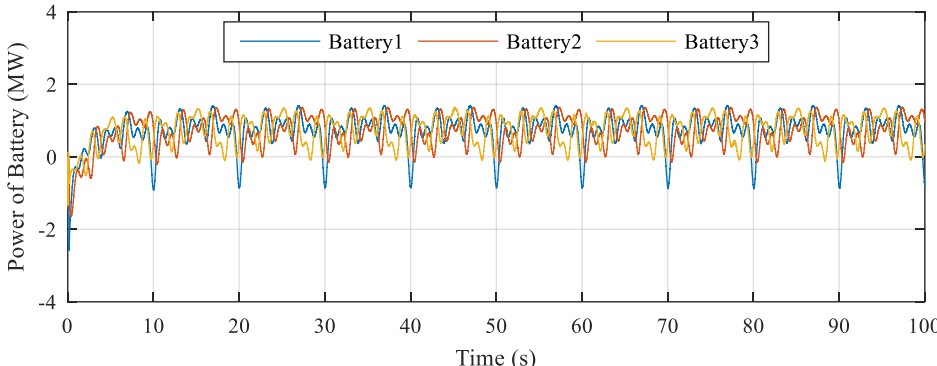

**Figure 11.** Power of battery without the proposed controller (Case 1).

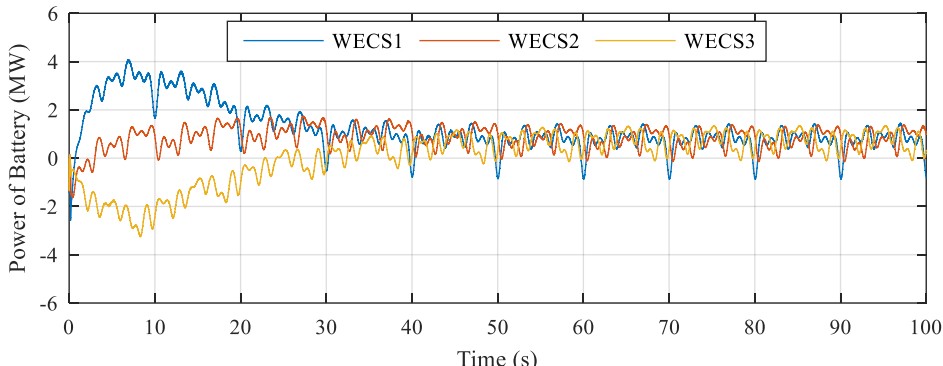

**Figure 12.** Power of battery with the proposed controller (Case 1).

### 4.2. Case Study 2: Performance of the Proposed Control Strategy with Same Initial SOCs

In the second case study, the effectiveness of the proposed controller is verified when the initial SOC of batteries is designed to have the same values. As can be seen from Figure 13, the SOC of batteries remains 50% at the beginning. Without the proposed controller, due to the different wind speeds of the WECSs, the SOCs slightly move far away from each other at the first interval of 20 s.

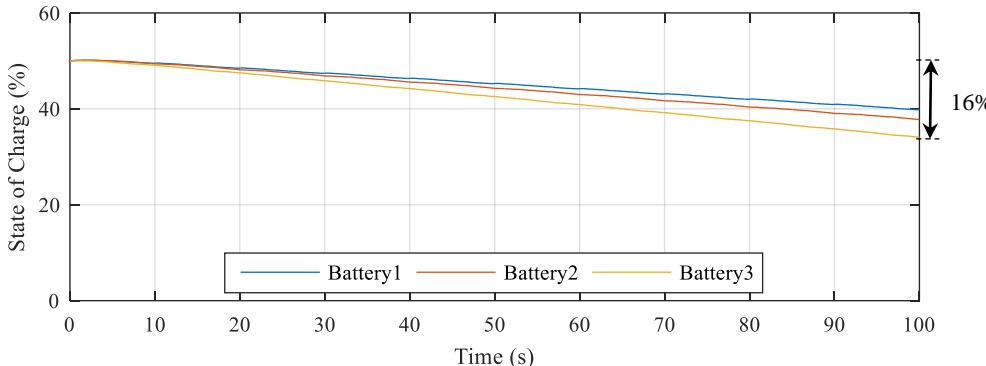

**Figure 13.** State of charge of batteries in the WECS without proposed controller (Case 2).

After that, it can be clearly seen that the SOCs are significantly different over the 100-s period. The SOC of battery 3 moves down quickly, and its slope is higher in comparison with other SOCs because battery 3 releases power the most. It reaches 34% at the end of the period, corresponding to 16% of power discharge.

On the other hand, Figure 14 shows the SOC of the batteries in case of applying consensus algorithm. From 50% at the initial time, the SOC of batteries moves down with a moderate slope, and they remain at a similar value until the end of the period. Moreover, the lowest amount of the SOC is 37%, corresponding to 13% of the power discharge. This value is smaller than the SOC of battery 3 without the consensus algorithm. Therefore, battery 3 releases less power to the GSC in this case.

In addition, Figures 15 and 16 illustrate the active output power flowing to the utility grid without the consensus algorithm and with the consensus algorithm, respectively. It is clear that in both cases the active power of GSCs is controlled to maintain at 1.5 MW. The power fluctuation is reduced by using a low-pass filter to damp out the high-frequency oscillation. For that reason, the active power of GSCs has small variation. In addition, without the proposed controller, the active power of GSCs are injected equally to the grid, while they have a small difference when the consensus algorithm is applied. It is because the SOCs of the batteries are equal that the batteries release the same power to the grid-side converter. Hence, with different wind speeds, the GSC of WECS1, which receives larger average power from the wind turbine, will transfer a little more power to the utility grid. By contrast, the output power of WECS3 is lower compared to the others because the average wind speed flowing

to WECS3 is the smallest. The battery power of both cases is similar because the average wind speed of WECSs is not much different, as depicted in Figures 17 and 18.

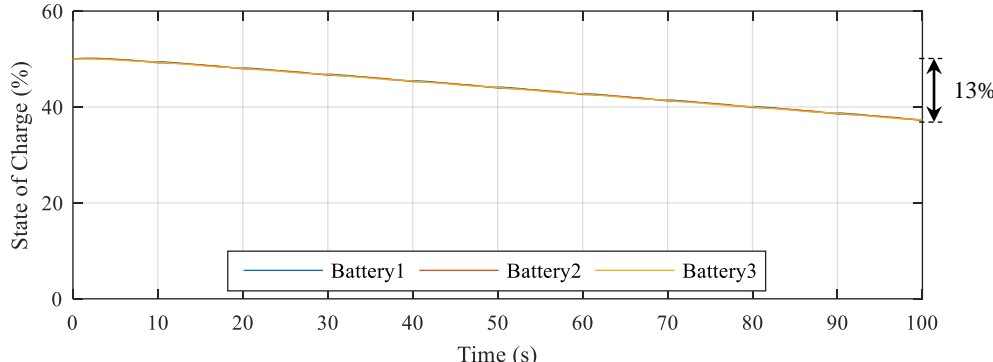

**Figure 14.** State of charge of batteries in the WECS with proposed controller (Case 2).

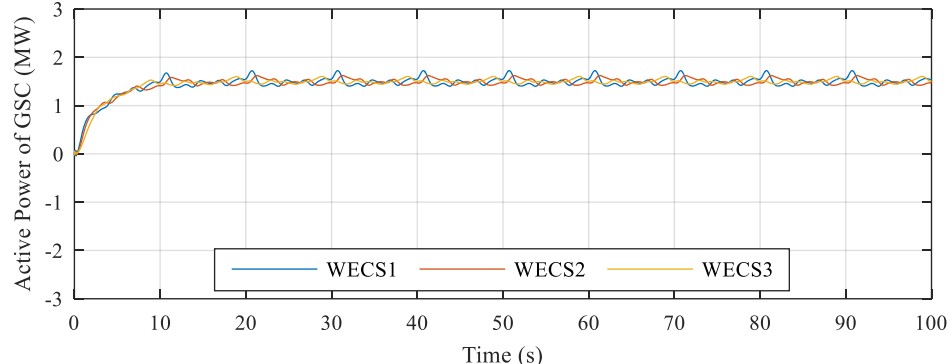

**Figure 15.** Active power of grid-side converter without proposed controller (Case 2).

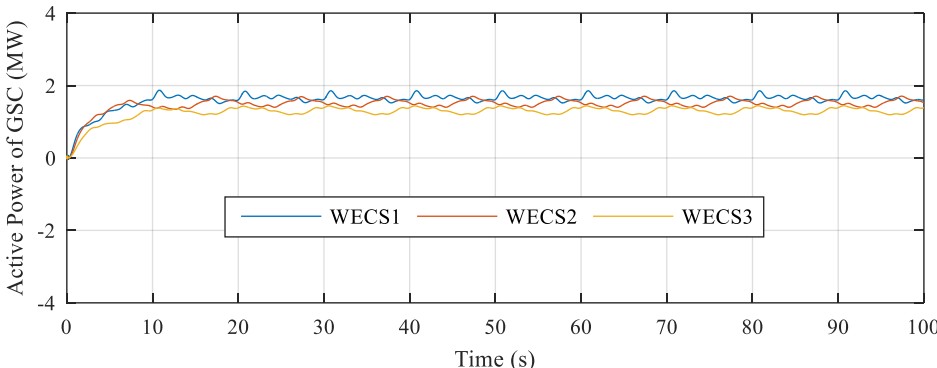

**Figure 16.** Active power of GSC with proposed controller (Case 2).

Furthermore, Figure 19 illustrates the total power of GSCs flowing into the utility grid in the case of a different initial SOC. The power fluctuation is smoothed because the fluctuation in wind power caused by the wind speed is avoided by means of the low pass filter in the GSC controller. There is a slight difference in the total power of the wind farm in the cases of with and without the proposed controller since two controllers consist of the strategy of smoothing wind power fluctuation.

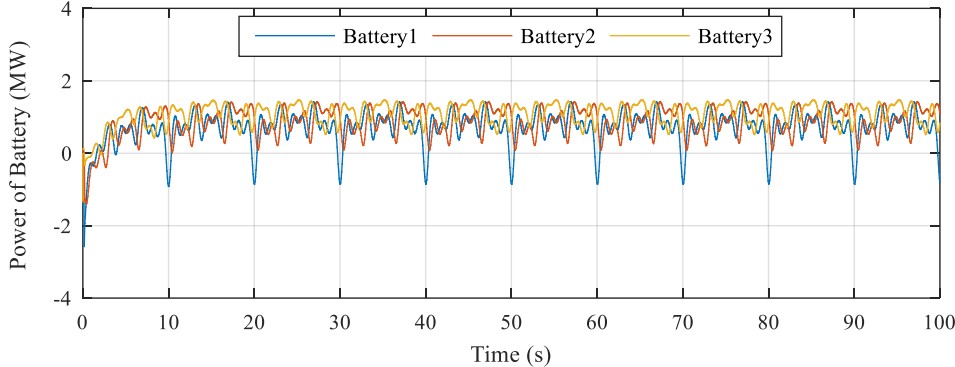

**Figure 17.** Power of battery without proposed controller (Case 2).

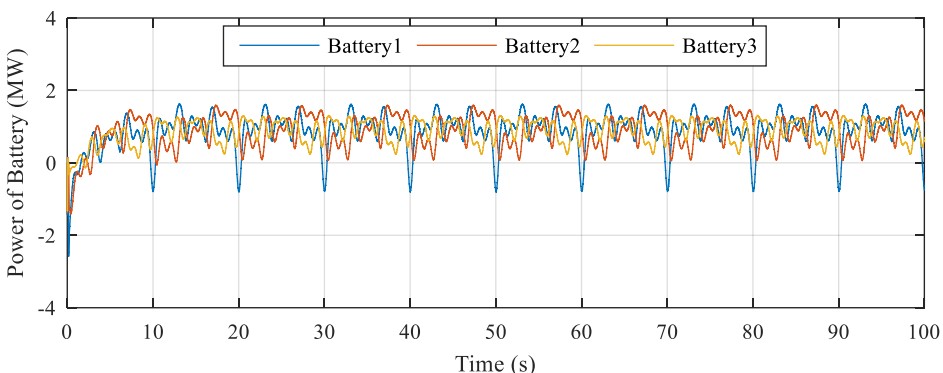

**Figure 18.** Power of battery with proposed controller (Case 2).

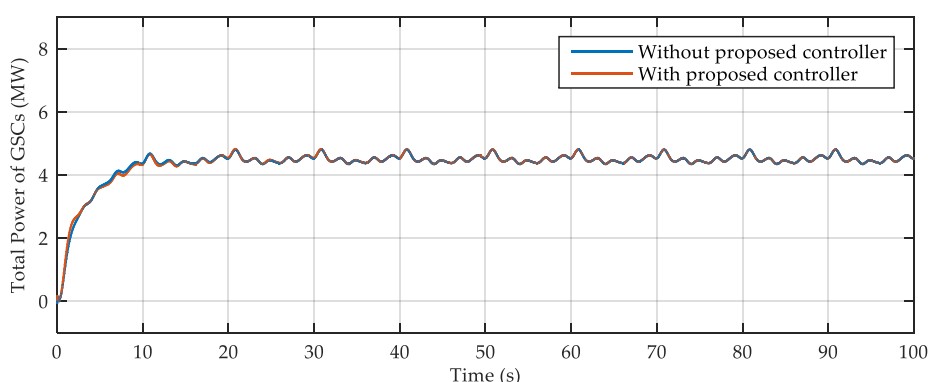

**Figure 19.** Total active power of GSCs in case of different initial SOCs.

## 5. Conclusions

In this paper, an improved VSG control based on consensus algorithm has been applied to the GSC for balancing the SOCs of batteries in the WECSs. The effectiveness of the proposed controller has been verified through different initial SOCs and same initial SOC conditions under different wind speeds. With the consensus-based SOC balancing controller, the batteries which have higher SOC could support the batteries which have lower SOC. Thus, this prevented the negative effects of the SOC imbalance on the operation of the WECS. In addition, the smoothing power control based on low pass filter has been applied to the GSC for compensating the active power fluctuation to the utility grid. Hence, beside the advantage of VSG to enhance the inertia to the WECS, the output power of the WECS has been smoother, while the SOC of batteries has been balanced.

**Author Contributions:** C.-K.N. conceived and designed the experiments; T.-T.N. and H.-J.Y. performed the experiments and analyzed the results; H.-M.K. revised and analyzed the results; C.-K.N. wrote the paper.

**Funding:** This research was supported by Korea Electric Power Corporation. (Grant number: R18XA03.)

**Conflicts of Interest:** The authors declare no conflict of interest.

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
