# Peer review of "Consensus-Based SOC Balancing of Battery Energy Storage Systems in Wind Farm"

_energies, doi:10.3390/en11123507_

Round 1
Reviewer 1 Report
In
this paper, an improved virtual synchronous generator control based on
consensus algorithm is applied to the grid-side converter for
balancing the stage of charge(SOCs)
of batteries in the wind energy conversion (WECSs). The effectiveness
of the proposed controller has been verified through different initial
SOCs and same initial conditions under different wind speeds. The
batteries which have higher SOC could support the batteries which have
lower SOC. Thus, this prevents the negative effects of the SOC imbalance
on operation of the WECS. The smoothing power control based on low
pass filter has been applied to the grid-side controller
(GSC) for compensating the active power fluctuation to the utility
grid. Hence, besides the advantage of virtual synchronous generator to
enhance the inertia to the WECS, the output power of the WECS has been
smoother while the SOC of batteries have been balanced. The effectiveness of the proposed control 26 strategy is validated through the simulation by using Matlab/Simulink environment. The presented results are innovative and the paper is well written with fine English language.
Author Response
We appreciate the positive remark of the reviewer.
Reviewer 2 Report
Authors present a simulation paper about Consensus-based SOC Balancing of Battery Energy Storage Systems in Wind Farm.
I have no special comments.
Author Response

(The authors gave the same response as above.)

Reviewer 3 Report
The manuscript described an improved virtual synchronous generator control based on consensus algorithm applied to the grid-side converter for balancing the SOCs of batteries in the WECSs. The effectiveness of the proposed controller has been verified through different initial SOCs and same initial SOCs conditions under different wind speeds. The design and simulation are well designed and presented. Suggest for publication in present form.
Author Response

(The authors gave the same response as above.)

Reviewer 4 Report
This paper introduces the consensus-based battery SOC balancing in the application of wind energy storage. The following comments are given for the author's consideration.
1. Please introduce the motivation of performing battery SOC balancing in BESS. If the battery packs for each WECS are independently operated, there is no need of SOC balancing.
2. In fact, according to Fig. 4, it is the total power of the three WECSs instead of the individual WECS power that should be regulated. Note that, it is the fluctuation of the total power of all WECS transferred to the grid that should be smoothed.
3. For the two study cases, give the figure for the total active power of the GSCs. It is important to compare the fluctuation of the total active power to the grid with and without the proposed control method.
4. Introduce how these PI parameters are determined or identified in Equations (6) and (7).
5. Label the BSC or BSC controller explicitly in Fig. 1 even though it is embedded in the BESS.
6. Some typos and grammatic errors are pointed out as below:
Line 33, which has been àwhich has
Line 44, capabilities àcapacities
Line 47, accelerating àaccelerated
Line 53, SOC-based on droop control àSOC based droop control
Line 58, derivation àdeviation
Line 105, … that make … à…, and thus, …
Line 127, Whereas àWhile
Line 142, from the BSC àby the BSC
Line 229, input àused as the input of
Line 259-260, rewrite the sentence starting from “By contrast, …”
Line 260, After 30 (s) àAfter 50 s
Line 310, besides àbeside
7. For the battery SOC balancing and its control, the following relevant papers are suggested to discuss for enhanced literature review:
[1] Estimation of Cell SOC Evolution and System Performance in Module-based Battery Charge Equalization Systems, IEEE Trans Smart Grid, 2018.
[2] Mathematical analysis and coordinated current allocation control in battery power module systems, Journal of Power Sources, 2017.
Author Response
The response to the reviewer's comments is attached.

Round 2
Reviewer 4 Report
All questions and suggestions have been well addressed.